

# In-hospital cardiac arrest (IHCA): survival status and its determinants in Malaysian public healthcare

Marhaini Mostapha[1], Mohd Shahri Bahari[1], Min Fui Wong[2], Sivaraj Raman[1], Farhana Aminuddin[1], Shaiful Jefri[1], Nur Amalina Zaimi[1], Nor Zam Azihan Mohd Hassan[1], Hin Kwang Goh[3], Chee Kin Yoon[4], Eric Tang[4], Meng Li Lee[4] and Lean Wah Luah[5,6]

[1] Centre for Health Economics Research, Institute for Health Systems Research, National Institute of Health, Ministry of Health, Shah Alam, Selangor, Malaysia
[2] Centre for Health Policy Research, Institute for Health Systems Research, National Institute for Health, Ministry of Health, Shah Alam, Selangor, Malaysia
[3] Director Office, Hospital Pulau Pinang, Ministry of Health, George Town, Pulau Pinang, Malaysia
[4] Medical Department, Hospital Pulau Pinang, Ministry of Health, George Town, Pulau Pinang, Malaysia
[5] Department of Anaesthesiology and Intensive Care, Hospital Pulau Pinang, Ministry of Health, George Town, Pulau Pinang, Malaysia
[6] Health Screening Centre, Hospital Lam Wah Ee, George Town, Pulau Pinang, Malaysia

Corresponding author
Min Fui Wong,
dr.estherwong@moh.gov.my

## ABSTRACT

**Background:** In-hospital cardiac arrest (IHCA) remains a significant clinical challenge despite advances in resuscitation and critical care. Enhanced inpatient monitoring and post-IHCA management have improved survival rates and better neurological outcomes at discharge. This study aims to evaluate the IHCA survival rate and analyse key determinants influencing survival status.

**Methods:** A cross-sectional study was conducted using retrospective secondary data from a northern referral tertiary public hospital's cardiac arrest registry, encompassing IHCA cases in patients aged 18 and above between February 1, 2018, and January 31, 2019. The data included patient demographics, clinical characteristics, IHCA event timing, return of spontaneous circulation (ROSC), survival status, and post-arrest neurological outcomes. Patient survival was measured from the initiation of resuscitation to discharge or death, with survival analysis performed. Factors associated with IHCA survival were explored using logistic regression.

**Results:** A total of 934 IHCA cases were analyzed. The mean patient age was 60.8 years, with most being male (63.9%) and of Chinese ethnicity (45.5%). IHCA commonly occurred in non-critical care areas (61.0%), with 79.6% admitted for medical conditions. Successful ROSC was achieved in 43.5% of cases, and 8.2% had a shockable first rhythm. Only 6.4% survived to discharge or 30-day, with 45% having good neurological outcomes.

**Conclusion:** The IHCA survival rate remains low despite advances in resuscitation. Key survival factors include arrest location, shockable rhythms, and CPR-adrenaline

dosage ratio. Strengthening early recognition, prompt intervention, and optimized post-arrest care may enhance IHCA outcomes, particularly in non-critical care areas.

## INTRODUCTION

In-hospital cardiac arrest (IHCA) is a life-threatening event that can occur in any hospitalized patient, requiring immediate medical intervention (*Andersen et al., 2019b*). It is characterized by the sudden loss of circulation within a hospital setting, necessitating resuscitation efforts such as chest compressions, defibrillation, or both (*Andersen et al., 2019b*). Unlike out-of-hospital cardiac arrest, IHCA occurs in a controlled environment where trained medical personnel and resuscitation equipment are readily available. However, despite these advantages, survival rates remain variable, largely influenced by the underlying cause of arrest, the timeliness of intervention, and the quality of post-resuscitation care. The causes of IHCA are multifaceted, often linked to preexisting cardiovascular disease (*Institute for Public Health (IPH) & Ministry of Health Malaysia, 2019*), respiratory failure, severe infections, electrolyte disturbances, medication-related adverse effects, or postoperative complications. These conditions can lead to hemodynamic instability and progressive deterioration, making early recognition and interventions essential (*Burtin et al., 2013*; *Pollack et al., 2018*; *Honarmand et al., 2018*).

Despite its significant impact on patient survival and morbidity, IHCA has received relatively limited attention compared to other major cardiovascular emergencies such as myocardial infarction, stroke, and out-of-hospital cardiac arrest (*Peberdy et al., 2003*; *Sinha et al., 2016*; *Andersen et al., 2019a*). This disparity in focus may be due to the perception that IHCA occurs within a hospital setting, leading to assumptions of better survival outcomes. However, global estimates suggest that IHCA affects approximately one to five patients per 1,000 hospital admissions, with mortality rates reaching up to 80%. (*Sandroni et al., 2007*; *Larkin et al., 2010*; *Holmberg et al., 2019*; *Alao et al., 2022*). Even among survivors, the consequences can be severe, with many experiencing long-term complications such as neurological deficits due to prolonged cerebral hypoxia. Cognitive impairment, physical disability, and reduced quality of life are common among those who do survive, placing a significant burden on healthcare systems and caregivers.

While the immediate availability of medical personnel and advanced equipment generally contributes to higher survival rates in better-equipped hospital settings, outcomes remain suboptimal, with survival typically ranging between 15% and 39% (*Andersen et al., 2019a*; *Virani et al., 2020*; *Gräsner et al., 2021*). This highlights the need for further improvements in both resuscitation practices and post-arrest care. Consequently, clinical practice guidelines emphasize the importance of rapid CPR initiation, the deployment of rapid response teams, and post-arrest interventions, alongside enhanced training for healthcare providers and increased intensive care capacity

(*Ministry of Health Malaysia, 2016*; *Hirlekar et al., 2017*; *Wu et al., 2021*). However, in resource-limited settings, particularly in low- and middle-income countries, strengthening basic life support (BLS) training, including CPR proficiency, is a crucial and cost-effective strategy for improving IHCA survival rates.

Despite the critical impact of IHCA on patient outcomes, there remains a significant gap in data collection and standardized reporting, particularly in Malaysia. The absence of a nationwide prospective IHCA registry limits policymakers' ability to develop effective strategies for improving survival rates and long-term patient outcomes. Additionally, existing studies primarily focus on high-income countries, making it difficult to extrapolate findings to local contexts. Given these gaps, this study aims to report the IHCA incidence and survival rates, and contributing factors in relation to the survival status, within a single tertiary public hospital's established IHCA database. The findings will serve as a foundation for the development of a standardized data collection and reporting system, ultimately guiding policy decisions and improving patient care nationwide.

## MATERIALS AND METHODS

### Setting and study design

This retrospective cross-sectional study utilized an internally built IHCA registry database from a tertiary public hospital in Peninsular Malaysia for the years 2018 to 2019. The facility was selected due to the availability of an advanced resuscitation team and as it also functions as the teaching and referral center for the northern region of Malaysia. This hospital is equipped with 1,163 beds and 29 operational general ICU beds. The IHCA database encompasses patient admission details, medical records, clinical data, resuscitation procedures, event timelines, and the patient's CPC status at discharge (Table S3).

### Population and sample

Based on proportions and effect sizes from similar studies, the maximum required sample size was estimated to be 1,230 using OpenEpi software, with a significance level of 0.05 and a statistical power of 0.8 (*Chen et al., 2016*; *Fernando et al., 2019*). Since the dataset included only 934 patients, universal sampling was employed. A *post-hoc* analysis using G-Power confirmed that a sample size of 934 was sufficient for this study.

### Inclusion and exclusion criteria

All IHCA patients aged 18 years and above, reported in the IHCA registry between February 1, 2018, and January 31, 2019, were included in this study. This analysis excluded OHCA patients who were successfully resuscitated before admission, those who underwent CPR concurrently with hospital admission, and those who did not receive resuscitation attempts during the study period. All individuals who met the inclusion and exclusion criteria, and experienced cardiac arrest at the study site and receiving care from the nearest healthcare provider (staff) were included.

## Data collection for IHCA registry

The primary treatment team initiated resuscitation efforts once the "Code-blue" calls, with ICU staff assisting when needed and all procedure adheres to guidelines for resuscitation (*Ministry of Health Malaysia, 2016*). After resuscitation, the team documented patient details, resuscitation methods, and cardiac arrest outcomes. Survivors were scheduled for follow-up at their outpatient clinics as standard practice. Data was collected from case report forms (CRFs) completed by the primary medical team at the study site. For missing data, a clinical research associate (CRA) reviewed CRFs, retrieved relevant information from patient case notes, and cross-validated it with the medical team before reporting to the IHCA registry. An automated validation program checked for data discrepancies, including invalid or out-of-range values. The process flow is illustrated in Fig. 1.

## Data management and validation

A data collection team was assigned by the core research team to monitor and ensure the quality of the data collection process from the registry department. The list of the selected samples was distributed among the enumerators and data was captured using a standardized data collection form. Variables used in this study were divided into three data domains-patient demographic details, clinical features, and post-arrest outcomes. The data domains and codebook for related data elements are shown in Table S1. There was no direct interaction between the investigators and the subjects as the data was provided by the resuscitating and primary teams.

## Statistical analyses

The data collected were entered into Microsoft Excel 365, cleaned, and then transferred to STATA 17.0 for analysis. The incidence rate of IHCA was calculated based on new IHCA cases per 100,000 hospital admissions. The dataset was examined for centrality, dispersion, normality, and missing data. Variables with < 5% missing data were included. Data visualization included histograms and quantile-quantile (QQ) plots for continuous variables and bar plots for categorical variables. Continuous data were reported as mean and standard deviation (SD), while categorical variables were presented as frequency and percentage. The chi-square or Fisher's exact test was used to assess differences between categorical variables and outcomes.

Logistic regression was performed to identify survival determinants among IHCA patients. Survivors were coded as "1" and deaths as "0." Explanatory variables included admission reason, admission time, IHCA location, initial cardiac rhythm, witnessed arrests, rescuer qualifications, total shocks, ROSC, adrenaline CPR ratio, total adrenaline administered, and CPR duration. These variables were coded as "1" (yes) and "0" (no). The model building followed the Hosmer-Lemeshow model development strategy. Variables with a $p$-value < 0.25 in univariate analysis were included in the full logistic regression model. Variables with $p$-values > 0.05 were removed using backward elimination, guided by the likelihood-ratio test. Effect modification of first-order interactions was assessed. The final model was evaluated for heteroscedasticity, linearity,

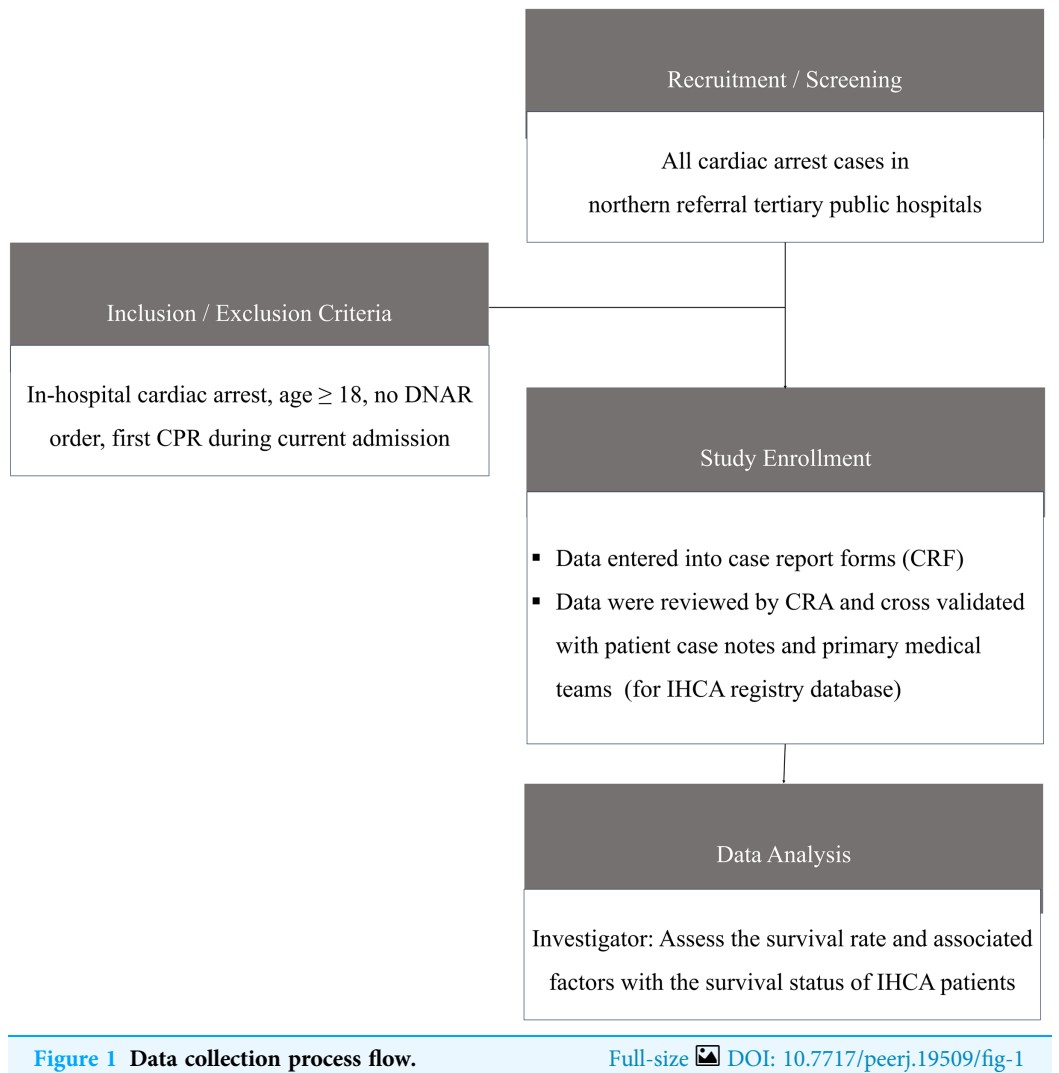

**Figure 1 Data collection process flow.**

goodness of fit, AUROC, and multicollinearity. A *p*-value < 0.05 was considered statistically significant.

Kaplan-Meier survival analysis was also applied to estimate the survival projections at key time points. This study examined survival rates from survival to discharge or 30 days following Utstein-reporting guidelines (*Nolan et al., 2019*). Survival time (T) was defined as the interval from resuscitation initiation to discharge or death. Survival status was coded as "0" (censored, non-death) and "1" (death, event). The Kaplan-Meier survival curve was plotted using Jamovi 2.3 for visualization. In this study, the data elements, outcomes of cardiac arrest, and cardiopulmonary resuscitation were referenced according to the Utstein resuscitation registry template (Table S1) for in-hospital cardiac arrest (*Nolan et al., 2019*).

## Ethics statement

This study was conducted according to the guidelines of the Declaration of Helsinki. This study was registered under the National Medical Research Register (NMRR), Ministry of
Health Malaysia (NMRR ID-22-02528-FV7). Informed consent was not required, as the data were obtained anonymously from the registry without patient identifiers.

## RESULTS

The overall incidence of adult IHCA was 19.8 per 1,000 hospital admissions. The demographic characteristics of the IHCA patients are presented in Table 1. The mean age of patients was 60.78 years (SD = 14.28). Patients were predominantly male (63.3%) and a majority of them were from the Chinese ethnicity (45.5%). Most of the patients (79.6%) were admitted for general medical services, while 11.8% were admitted for emergency surgery, and 6.6% were due to trauma. Interestingly, 69.8% of cases of IHCA happen after office hours. It was also found that the highest IHCA cases occurred in non-critical care areas such as the general wards (61.0%) compared to critical care areas which include intensive care units (ICU), high dependency units (HDU), and high dependency wards (HDW) and emergency department.

About 89.5% of IHCA cases had bystander witnesses, and most of them were rescued by medical officers (70.7%). Out of these attending medical officers, 81.8% of them were ALS-trained. Approximately 8.2% of their first documented cardiac rhythms were shockable while 91.8% were non-shockable. As for the CPR to adrenaline ratio, 85.4% of IHCA cases had durations exceeding 3 min/1 mg dosage. This led to a successful ROSC in 43.5% of cases, with an average CPR duration of 26 min. Among the survivors who received post-resuscitation care, 51.2% were managed in critical care units, including the ICU, HDU, and HDW, while the remaining 48.8% received treatment exclusively in general wards.

Among 6.4% (n = 60) of the survivors from IHCA, 45% (n = 27) displayed favorable neurological outcomes with CPC scores of 1 and 2 at discharge. Conversely, the remaining cases were categorized as CPC 3 and 4 at discharge (refer to Table S3 for the CPC scale). From Fig. 2, it is evident that the deaths among IHCA cases were primarily attributed to septicemia (45.6%), followed by cardiovascular disease (29.5%), and gastrointestinal/hepatobiliary (4.1%).

### In-hospital cardiac arrest survival status

The Kaplan-Meier survival plot (Fig. 3) and Table 2 show a rapid decline in IHCA survival, dropping by 84.6%, leaving only 15.4% (95% CI [13.28–17.96]) of survivors within the first 24 h post-resuscitation. The survival rate declines to 8.4% by day 7, 3.6% by day 30, and 2.7% by day 60, with an overall median survival time of 0.80 h (95% CI [0.7–0.8]).

### Factors associated with survival of IHCA

Nine variables with $p < 0.25$ from the univariate analysis (Table 3) were included in the final logistic regression model. These variables were the location of the IHCA event, first documented cardiac rhythm, arrest witnesses, CPR duration: adrenaline dosage ratio, age, ROSC, resuscitation duration (minutes), rescuer ALS training, and rescuer designation. Among these, the final logistic regression analysis (Table 4) identified three significant

**Table 1 Demographic, clinical characteristics and outcomes of IHCA patients (_n_ = 934).**

| Characteristic | | Total | | Missingness |
|---|---|---|---|---|
| | | **_n_** | **%** | **% (_n_)** |
| Age | Mean ± SD | Mean: 60.8 ± 14.3 | | 0.3(3) |
| | Median | Median: 63 years | | |
| | | Min:18 years | | |
| | | Max: 98 years | | |
| Gender | Male | 591 | 63.3 | 0 |
| | Female | 343 | 36.7 | |
| Race | Malay | 328 | 35.1 | 0 |
| | Chinese | 425 | 45.5 | |
| | Indian | 152 | 16.2 | 0 |
| | Others | 29 | 3.1 | |
| Time of IHCA event occurs | During office hours | 282 | 30.2 | 0 |
| | After office hours | 652 | 69.8 | |
| Reasons admission | Elective surgery | 16 | 1.7 | 0 |
| | Emergency surgery | 110 | 11.8 | |
| | Medical | 744 | 79.6 | |
| | Trauma | 62 | 6.6 | |
| | Others | 3 | 0.3 | |
| Location of IHCA event occurs | Non-critical care | 570 | 61.0 | 0 |
| | Critical care area | 364 | 39.0 | |
| First documented cardiac rhythm | Non-shockable | 857 | 91.8 | 0 |
| | Shockable | 77 | 8.2 | |
| Arrest witnessed | No | 98 | 10.5 | 0 |
| | Yes | 836 | 89.5 | |
| Rescuer ALS-Trained | No | 167 | 18.2 | 1.6 (15) |
| | Yes | 752 | 81.8 | |
| Rescuer designation | Medical officer | 660 | 70.7 | 0 |
| | House officer | 60 | 6.4 | |
| | Specialist | 86 | 9.2 | |
| | Others | 128 | 13.7 | |
| CPR duration: adrenaline dosage ratio | >3 min/1 mg | 798 | 85.4 | 0 |
| | ≤3 min/1 mg | 136 | 14.6 | |
| ROSC | Yes | 399 | 43.5 | 0 |
| | No | 535 | 56.5 | |
| Survival status | Survived to Discharge or at 30-day | 60 | 6.4 | 0 |
| | Dead | 874 | 93.6 | |

factors ($p < 0.05$) positively associated with IHCA survival: IHCA occurring in critical care areas (ICU, HDU, HDW, and ED), a shockable first documented cardiac rhythm, and a CPR duration: adrenaline dosage ratio of ≤ 3 min per 1 mg of adrenaline administered. Arrest witnesses remained a key factor for model fit and stability. The model showed a

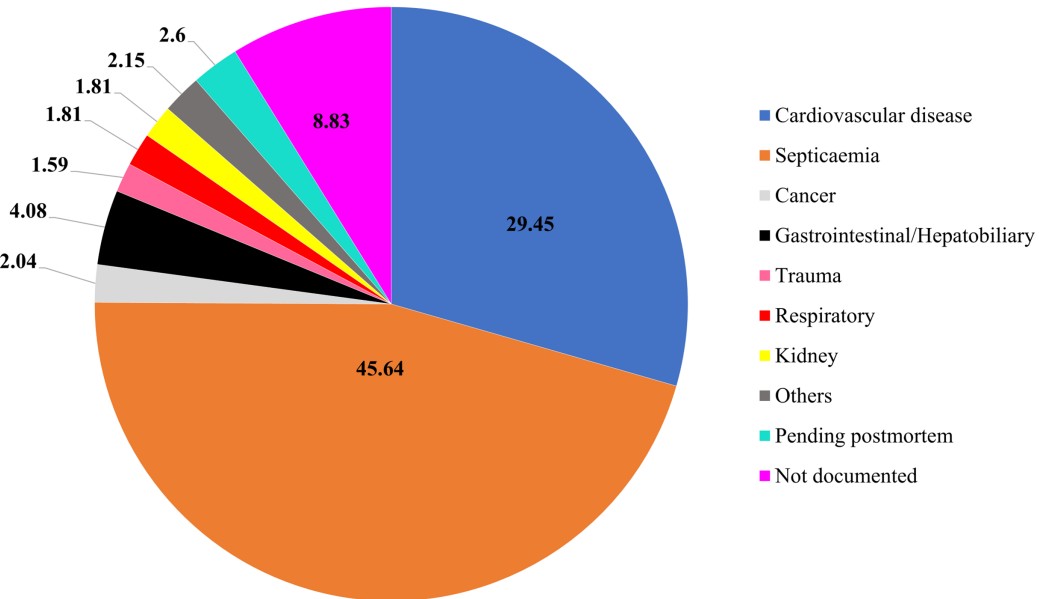

**Figure 2 Causes of death for IHCA patients (%).**

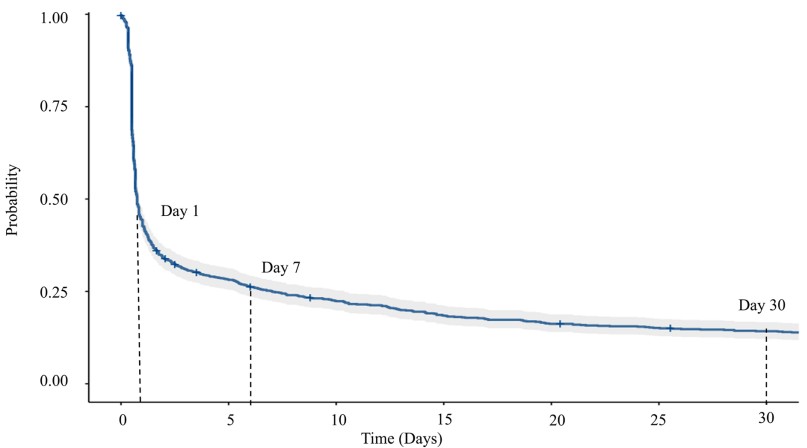

**Figure 3 Survival plot based on Kaplan-Meier estimate for overall IHCA cases.**

**Table 2 The survival rate with relevant time points.**

| Time (Hours) | Day | Number at risk | Number of events | Survival (%) | 95% confidence interval | |
|---|---|---|---|---|---|---|
| | | | | | Lower | Upper |
| 24 | 1 | 141 | 786 | 15.4 | 13.3 | 18.0 |
| 168 | 7 | 63 | 60 | 8.4 | 6.7 | 10.4 |
| 720 | 30 | 12 | 26 | 3.6 | 2.3 | 5.5 |
| 1,440 | 60 | 3 | 1 | 2.7 | 1.3 | 5.5 |

**Table 3 Univariate logistic regression of IHCA patients in relation to survival (n = 934).**

| Variables | | Survived n (%) | Death n (%) | Crude odds ratio (95% CI) | Chi-square | p-value |
|---|---|---|---|---|---|---|
| Age | | | | 0.99 [0.97–1.01] | – | 0.210* |
| Gender | Male | 37 (6.6) | 554 (93.4) | Reference | | |
| | Female | 21 (6.1) | 322 (93.9) | 0.92 [0.53–1.60] | 0.082 | 0.775 |
| Race | Malay | 24 (7.3) | 304 (92.7) | Reference | | |
| | Chinese | 24 (5.7) | 401 (94.4) | 0.76 [0.42–1.36] | 0.875 | 0.354 |
| | Indian/Others | 12 (6.6) | 169 (93.4) | 0.99 [0.44–1.84] | | 0.772 |
| Time of IHCA event occurs | Office Hours | 16 (5.7) | 266 (94.3) | Reference | | |
| | After Office Hours | 44 (6.8) | 608 (93.3) | 1.20 [0.67–2.17] | 0.378 | 0.539 |
| Reason admission | Non-Medical | 11 (5.8) | 179 (94.2) | Reference | | |
| | Medical | 49 (6.6) | 695 (93.4) | 1.15 [0.58–2.25] | 0.1597 | 0.890 |
| Location of IHCA event occurs | Non-Critical care | 23 (3.9) | 568 (96.1) | Reference | | |
| | Critical care | 37 (10.8) | 306 (89.2) | 2.99 [1.74–5.12] | 17.167 | <0.001*** |
| First documented cardiac rhythm | Non-Shockable | 43 (5.0) | 814 (95.0) | Reference | | |
| | Shockable | 17 (22.1) | 60 (77.9) | 5.36 [2.87–9.97] | 34.209 | <0.001*** |
| Arrest witnessed | No | 3 (3.1) | 95 (96.9) | Reference | | |
| | Yes | 57 (6.8) | 779 (93.2) | 2.15 [0.71–7.54] | 2.059 | 0.163* |
| Rescuer ALS-trained | Yes | 52 (6.9) | 700 (93.1) | Reference | | |
| | No | 7 (4.2) | 160 (95.8) | 0.59 [0.93–6.00] | 1.687 | 0.199* |
| Rescuer designation | Medical Officer | 45 (6.8) | 615 (93.2) | Reference | | |
| | House Officer | 4 (6.7) | 56 (93.3) | 0.98 [0.34–2.81] | 1.570 | 0.964 |
| | Specialist | 6 (7.0) | 80 (93.0) | 1.03 [0.42–2.48] | | 0.956 |
| | Others | 5 (3.9) | 123 (96.1) | 0.56 [0.23–1.42] | | 0.222* |
| Resuscitation duration (minutes) | | | | 0.90 [0.88–0.93] | 0.460 | <0.001*** |
| CPR duration: adrenaline dosage ratio | >3 min/1 mg | 36 (4.5) | 762 (95.5) | Reference | | |
| | ≤3 min/1 mg | 24 (17.7) | 112 (82.4) | 4.53 [2.61–7.89] | 33.353 | <0.001*** |
| ROSC | Yes | 56 (14.1) | 342 (85.9) | Reference | | |
| | No | 1 (0.0) | 535 (99.8) | | 80.083 | <0.001*** |

**Note:**
$p < 0.001$*** , $p < 0.01$** , $p < 0.25$* significant; ALS, Advance life support; ROSC, Return of Spontaneous Circulation; CPR, Cardiopulmonary Resuscitation; IHCA, In-hospital cardiac arrest.

**Table 4 Multi-variables logistic regression for IHCA patients in relation to survival (n = 934).**

| Variables | | Crude OR (95% CI) | p-value | Adjusted OR (95% CI) | p-value |
|---|---|---|---|---|---|
| Location of IHCA event occurs | Non-critical care | Reference | | Reference | |
| | Critical care | 2.99 [1.74–5.12] | <0.001*** | 1.94 [1.08–3.47] | 0.026* |
| First documented cardiac rhythm | Non-Shockable | Reference | | Reference | |
| | Shockable | 5.36 [2.87–9.97] | <0.001*** | 3.98 [2.05–7.71] | <0.001*** |
| Arrest witnessed | No | Reference | | Reference | |
| | Yes | 2.15 [0.71–7.54] | 0.163 | 1.31 [0.39–4.42] | 0.667 |
| CPR duration: adrenaline dosage ratio | >3 min/1 mg | Reference | | Reference | |
| | ≤3 min/1 mg | 4.53 [2.61–7.89] | <0.001*** | 3.66 [2.05–6.55] | <0.001** |

(Continued)

| Variables | | Crude OR (95% CI) | p-value | Adjusted OR (95% CI) | p-value |
|---|---|---|---|---|---|
| Age | | 0.99 [0.97–1.01] | 0.210 | – | – |
| ROSC | Yes | | | | |
| | No | 80.08 | <0.001*** | – | – |
| Resuscitation duration (minutes) | | 0.90 [0.88–0.93] | <0.001*** | – | – |
| Rescuer ALS-trained | Yes | Reference | 0.199 | – | – |
| | No | 0.59 [0.93–6.00] | | | |
| Rescuer designation | Medical officer | Reference | | – | – |
| | House officer | 0.98 [0.34–2.81] | 0.964 | | |
| | Specialist | 1.03 [0.42–2.48] | 0.956 | | |
| | Others | 0.56 [0.23–1.42] | 0.222 | | |

**Note:**
$p < 0.05$ mean VIF = 1.04 AUC = 0.749 (95% CI [0.692–0.830]), goodness-of-fit-test = Prob > chi$^2$ = 0.678 ($p > 0.05$). Classification table: 94%, McFadden R$^2$= 0.114 ($p < 0.001^{***}$, $p < 0.01^{**}$, $p < 0.05^*$)

**Table 5 Comparison of ROSC, survive to discharge among different countries.**

| Country | Incident rate of IHCA (per 1,000 admissions) | ROSC (%) | Survived to discharge (%) | CPC score (%) | References |
|---|---|---|---|---|---|
| Malaysia | 19.80 | 43.5 | 6.4 | CPC 1,2: 45.0 | This study |
| China | 17.50 | 35.5 | 9.1 | CPC 1,2: 6.4 | Shao et al. (2016) |
| Taiwan | 3.25 | 66.0 | 11.8 | CPC 1,2: 46.7 | Chen et al. (2016) |
| Singapore | 1.03 | 90.0 | 10.0 | N/A | Lyu et al. (2021) |
| United Kingdom | 1.60 | 45.0 | 18.4 | CPC 1,2: 97.5 | Nolan et al. (2014) |
| Italy | 1.51 | 35.7 | 14.8 | CPC 1,2: 90.8 | Radeschi et al. (2017) |
| Uganda | Not mentioned | 62.2 | 14.9 | CPC 1,2: 12.2 | Alum et al. (2022) |

good fit, with multicollinearity excluded for all variables, including arrest witnesses, as shown in Table 4.

# DISCUSSION

At 19.8 per 1,000 admissions, the incidence of IHCA in this study was the highest reported among selected studies worldwide (Nolan et al., 2014; Chen et al., 2016; Shao et al., 2016; Radeschi et al., 2017; Lyu et al., 2021; Alum et al., 2022) (Table 5). This notably high incidence may be attributed to differences in patient populations within the study setting. The key finding was the high prevalence of non-shockable rhythms, such as asystole and pulseless electrical activity in our sample. These incidences are often associated with poorer prognosis compared to shockable rhythms like ventricular fibrillation or pulseless ventricular tachycardia. Additionally, septicemia emerged as the leading cause of death, further emphasizing the role of severe infections and multi-organ dysfunction in IHCA outcomes (Fig. 2). In contrast, other studies have reported lower rates of non-shockable rhythms, with cardiovascular conditions such as acute myocardial infarction and heart failure being the predominant comorbidities associated with IHCA (Chen et al., 2016; Shao

*et al., 2016*). This discrepancy suggests potential differences in patient demographics, underlying health conditions, and pre-arrest care, all of which can significantly influence the outcomes of IHCA.

Despite a relatively high return of spontaneous circulation (ROSC) rate of 43.5%, the survival-to-discharge rate remained low at 6.4%. Notably, the ROSC rate observed in this study is comparable to those reported in developed countries such as the United Kingdom (*Nolan et al., 2014*), Italy (*Radeschi et al., 2017*), and China (*Shao et al., 2016*) (Table 5), reflecting effective early resuscitation efforts. While achieving the return of spontaneous circulation (ROSC) is a critical first step in in-hospital cardiac arrest (IHCA) management, the low survival rates and high incidence of sepsis observed in this study highlight the need to re-evaluate post-resuscitation care. Optimizing intensive care strategies, structured neurological monitoring, and early rehabilitation interventions can be considered to mitigate post-resuscitation complications like multi-organ failure to enhance long-term patient outcomes. Strengthening local hospital protocols and healthcare provider training in post-resuscitation management should also be implemented.

## Determinants of IHCA outcomes

In this study, the shockability of the initial cardiac rhythm emerged as a key independent factor influencing survival among IHCA cases. These findings were supported by studies that observed the positive relationship between the survival rates and the presence of shockable rhythms such as ventricular tachycardia (VT) and ventricular fibrillation (VF) (*Fennessy et al., 2016*; *Aziz et al., 2018*). Timely defibrillation significantly improves survival outcomes in these cases (*Andersen et al., 2019a*). However, a study from the USA reaffirms that survival is not solely dependent on rhythm shockability but also on the effectiveness of acute resuscitation efforts and post-resuscitation care (*Girotra et al., 2012*). This underscores the need for a comprehensive approach to IHCA management that extends beyond initial rhythm assessment.

A high proportion of non-shockable rhythms (91.8%) was observed in this study, similar to findings from Abu Dhabi (91.1%) (*Aziz et al., 2018*) and Turkey (87.9%) (*Yilmaz & Omurlu, 2019*) but higher than reports from other countries. According to Denmark's IHCA registry, non-shockable rhythms are often linked to non-cardiovascular conditions, particularly sepsis (*Stankovic et al., 2021*). Given the high proportion of sepsis in our study cohort (Fig. 2), this could explain the predominance of non-shockable rhythms. These findings emphasize the importance of early sepsis detection and management in preventing the deterioration of IHCA (*Yakar et al., 2022*). Studies indicate that public awareness of sepsis remains low (*Parsons Leigh et al., 2022*), yet targeted sepsis education has been shown to improve recognition and early intervention (*Fiest et al., 2022*). In resource-limited settings, healthcare providers could leverage opportunistic education strategies during routine consultations to inform high-risk patients about sepsis symptoms and timely medical intervention.

Another critical observation in this study was the lower-than-recommended adrenaline administration interval, with infusion occurring in less than 3 min, despite protocol recommendations of three to 5 min (*American Heart Association, 2000*). While shorter

total resuscitation durations (<20 min) and adequate adrenaline dosages were observed, a high proportion of unfavorable neurological outcomes were also noted upon discharge. This raises concerns about the potential neurological impact of early adrenaline administration. Observational data suggest that while earlier adrenaline use may improve ROSC, administration within 2 min of the first defibrillation shock may be detrimental (*Gough & Nolan, 2018*). A large trial funded by the National Institute for Health and Care Research (NIHR) found that although adrenaline increased 30-day survival, it was associated with a higher risk of severe brain damage (*Perkins et al., 2018*). Similarly, other studies indicate that adrenaline does not necessarily correlate with long-term favorable neurological outcomes (*Ludwin et al., 2021*). Given these uncertainties, regular audits and follow-up studies on adrenaline and other clinical parameters used during resuscitation are necessary to refine their role in IHCA management. Based on the study findings, IHCA recognition and survival may be improved through effective early warning systems, continuous monitoring, and rapid response activation. High-quality CPR and early defibrillation are critical, supported by training and real-time feedback. A well-coordinated Code Blue team with clear roles enhances response efficiency. Post-resuscitation care, including temperature management and hemodynamic support, aids patient recovery.

## Policy implication

Our findings underscore the significant association between the location of cardiac arrest and IHCA survival rates. Patients who experienced cardiac arrest in monitored areas, such as the ICU, Cardiac Care Unit (CCU), and emergency care units, had notably higher survival rates than those in general wards (*Sandroni et al., 2004*). This suggests that continuous monitoring and rapid intervention play a protective role, even in critically ill patients. Similarly, our study highlights the importance of the location of IHCA events, as areas with advanced monitoring such as the ICU, CCU, and emergency units showed higher IHCA survival rates. This is due to the readily available monitoring system, rapid access to resuscitation equipment, trained staff, and heightened vigilance, enabling faster intervention and better critical care management (*Perman et al., 2016*; *Vo et al., 2024*). To improve survival in general wards and facilities where ICU beds are limited, a systematic approach to patient bed allocation is necessary (*Sandroni et al., 2004*; *Pangan et al., 2021*). Identifying high-risk individuals and ensuring they receive enhanced monitoring and immediate resuscitation support can mitigate delays in advanced life support (ALS). However, the absence of local risk stratification guidelines complicates this process (*Starks et al., 2018*). Developing a comprehensive IHCA registry tailored to the Malaysian healthcare system would help identify at-risk patients and inform evidence-based preventive strategies (*Chan et al., 2023*; *Vo et al., 2024*).

Beyond initial resuscitation efforts, post-resuscitation care plays a crucial role in determining survival outcomes. Our study found that IHCA location, witnessed arrests, shockable rhythms, and CPR adrenaline duration independently influenced patient survival. This suggests that factors beyond rapid defibrillation, such as targeted temperature management, arterial line monitoring, perfusion maintenance, and early cardiac catheterization, are essential components of IHCA management (*Nolan et al.,*

*2015*). These interventions are more likely to be available in ICU settings than in general wards, reinforcing the critical role of intensive care services in improving IHCA outcomes. Therefore, strengthening post-resuscitation protocols and ensuring their accessibility beyond ICU settings should be a priority.

The disparity between high ROSC rates and low survival-to-discharge rates in our study highlights the urgent need to expand intensive care capacity. A 2015 report indicated that Malaysian public hospitals had a median ICU bed occupancy rate of 91.6%, with approximately 32% of patients denied ICU admission due to bed shortages (*Ling et al., 2015*). Our study hospital reported ICU occupancy rates ranging from 89.0% to 94.0%, further emphasizing the resource strain. To improve IHCA survival, expanding ICU infrastructure and optimizing resource allocation are critical. In the short term, deploying additional critical care personnel, advanced monitoring systems, and resuscitation equipment to high-risk areas such as general wards can help bridge the gap in post-resuscitation care.

### Study limitations

A key limitation of this pioneering study, despite its substantial sample size of 934 cases, is its single-center design. This restricts the ability to generalize findings across the broader Malaysian healthcare system. While the study provides valuable insights into IHCA incidence, survival rates, and associated factors, it does not account for several critical determinants that could influence patient outcomes. Notably, comorbidities, post-resuscitation vital signs, delays before the first rhythm assessment, certain laboratory parameters, and medication usage were not analyzed due to registry data limitations. These factors are essential in understanding the full spectrum of IHCA predictors and interventions, and their exclusion may limit the study's ability to offer a comprehensive assessment of survival determinants. Thus, it is recommended that the relevant specialized department improve the IHCA registry, ensuring compliance with the Utstein resuscitation registry template for in-hospital cardiac arrest reporting guidelines (Table S1). Future research incorporating multicenter data and a more extensive range of clinical variables would be beneficial in refining IHCA management and policy strategies.

## CONCLUSIONS

In-hospital cardiac arrest remains a challenge, with low survival-to-discharge rates despite moderate ROSC success, which is associated with arrest location, shockable rhythms, and CPR duration to adrenaline dosage ratio. Enhancing post-arrest care, expanding ICU capacity, and establishing a robust national IHCA registry for comprehensive data collection and monitoring are vital steps toward improving outcomes.

## ACKNOWLEDGEMENTS

The authors would like to thank the Director-General of Health Malaysia for granting permission to publish this article. We also gratefully acknowledge the staff from the Department of Anaesthesiology & Intensive Care, Hospital Management, and the Department of Medicine, Hospital Pulau Pinang, Malaysia, as well as the Institute for

Health Systems Research, National Institute of Health Malaysia. ChatGPT was used in this work in order to improve the readability and language of the work. After using this tool, the authors reviewed and edited the content as needed and takes full responsibility for the content of the publication.

### Funding
The authors received no funding for this work.

### Competing Interests
The authors declare that they have no competing interests.

### Author Contributions
- Marhaini Mostapha conceived and designed the experiments, performed the experiments, analyzed the data, prepared figures and/or tables, authored or reviewed drafts of the article, and approved the final draft.
- Mohd Shahri Bahari conceived and designed the experiments, performed the experiments, analyzed the data, prepared figures and/or tables, and approved the final draft.
- Min Fui Wong conceived and designed the experiments, performed the experiments, analyzed the data, prepared figures and/or tables, authored or reviewed drafts of the article, and approved the final draft.
- Sivaraj Raman performed the experiments, analyzed the data, prepared figures and/or tables, authored or reviewed drafts of the article, and approved the final draft.
- Farhana Aminuddin performed the experiments, prepared figures and/or tables, and approved the final draft.
- Shaiful Jefri performed the experiments, prepared figures and/or tables, and approved the final draft.
- Nur Amalina Zaimi performed the experiments, prepared figures and/or tables, and approved the final draft.
- Nor Zam Azihan Mohd Hassan performed the experiments, prepared figures and/or tables, and approved the final draft.
- Hin Kwang Goh performed the experiments, prepared figures and/or tables, and approved the final draft.
- Chee Kin Yoon performed the experiments, prepared figures and/or tables, and approved the final draft.
- Eric Tang performed the experiments, prepared figures and/or tables, and approved the final draft.
- Meng Li Lee performed the experiments, prepared figures and/or tables, and approved the final draft.
- Lean Wah Luah performed the experiments, prepared figures and/or tables, and approved the final draft.

## Human Ethics

The following information was supplied relating to ethical approvals (*i.e.*, approving body and any reference numbers):

Medical Research and Ethics Committee (MREC) of the Ministry of Health (MOH) Malaysia (No. 22-02528-FV7(2)).

## Ethics

The following information was supplied relating to ethical approvals (*i.e.*, approving body and any reference numbers):

Medical Research and Ethics Committee (MREC) of the Ministry of Health (MOH) Malaysia (No. 22-02528-FV7(2)).

## Data Availability

The code and raw data are available in the Supplemental Files.

## Supplemental Information

Supplemental information for this article can be found online at http://dx.doi.org/10.7717/peerj.19509#supplemental-information.

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
