# Peer review of "In-hospital cardiac arrest (IHCA): survival status and its determinants in Malaysian public healthcare"

_PeerJ, doi:10.7717/peerj.19509_

## Round 0.1 · original submission · Major Revisions

Dear authors,

Thank you for your submission to PeerJ. After revision, some main concerns were raised that need to be addressed before publication. For example, it is suggested (and relevant) to add laboratory parameters, troponin levels, and the proportion of patients who had coronary angiograms. Also, to include a brief description of hospital activity (e.g., admissions, ED attendances, surgical procedures) and infrastructure (e.g., training standards, cardiac arrest teams, defibrillators, early warning systems). Other key points include:

- Clarify data validation processes and how IHCA incidence was routinely collected.
- Clearly articulate the study aim and address any grammatical ambiguities.
- Explain the purpose of Table 2.
- Provide details on variables used in multivariable logistic regression (e.g., in Table 4).
- Include median age and other descriptive data (e.g., admission diagnoses, co-morbidities, prescribed medicines, surgical procedures).
- Explain the finding that "witnessed arrest" is not significant and address potential collinearity issues.
- Describe elapsed time before first rhythm monitoring and provide details on time delays (e.g., to first adrenaline use).
- Ensure the study includes all 934 IHCA cases reported and clarify how missing data were minimized.
- Align the conclusion with the study results.
- Discuss potential next steps for research locally and nationally, and ensure the discussion includes updated citations.
- Also, consider to revise the color coding in Figure 1 to avoid confusion caused by multiple blue tones.

Looking forward to receiving the revised manuscript.

·

Basic reporting

The article is well-written and indicates the prognostic parameters.
In non-shockable rhythms, the basic rhythm observed and an ECG finding later after resuscitation could have been informative.
The majority of non-shockable rhythms were in the context of sepsis. An idea regarding electrolytes, particularly potassium and renal function, would have added more value.

Experimental design

No comment

Validity of the findings

No comment

Additional comments

If the references could have been numbered, it would have been better.
What needs to be added is the laboratory parameters, troponin levels and proportion of patients who had coronary angiograms.

·

Basic reporting

An interesting and potentially important descriptive study, exploring IHCA in one Malaysian hospital. There are small numbers of grammatical errors, full English language proofing required.

The introduction is quite long and could be more concise.

Many of the references are quite old and while older descriptive material can be useful, current science, standards and performance issues should use up to date references.

Please provide a brief description of hospital activity e.g. number of admissions, ED attendances, surgical procedures during the year. What training structures and standards are in place for staff and how are they monitored? Are cardiac arrest teams/crash carts/defibrillators widely available within what appears to be a very large institution? Is any form of early warning scoring system in use? While CPR and defibrillation are fundamentals, it is well recognised that IHCA includes a high proportion of non-shockable rhythms and a full spectrum of diagnostic and management strategies should be available.

Experimental design

A retrospective, descriptive study based on what appears to be a bespoke registry and data collection system within the hospital. It's unclear how the data were validated and how incidence of IHCA was routinely collected - please clarify.

The study aim at the end of the introduction is quite unclear as it refers to ?quality improvement efforts included in the study. No evidence of QI interventions are reported here and the study design does not have a before/after structure to report in impact. I suspect this is a grammatical rather than a design issue - please clarify.

The extensive statistical analysis seems thorough but it's unclear what the purpose of Table 2 is - please clarify.

Percentage data should be reported to only two places of decimals, rather than three, for relatively small numbers.

Validity of the findings

The key findings here are the very high incidence of IHCA and the very low survival rates. The very high proportion of non-shockable rhythms also raise the issue of elapsed time before first rhythm monitoring - no time delays are reported, apart from ?time to first adrenaline use.

It's unclear if the 934 cases reported are all IHCAs which occurred during the year - how was the registry maintained to ensure this happened? The lowlevel of missing data is impressive.

A median age would be useful, given the wide range.

The data are clear and describe key elements but do not include any of the more complex descriptors which might guide further research e.g. admission diagnosis, co-morbidities, prescribed medicines, surgical procedures etc.

Additional comments

The discussion covers a range of relevant issues and makes some useful suggestions about how recognition and survival from IHCA might be improved - again, some up to date citations might be of value. It would also be useful for the authors to explore next research steps in this important area - while this baseline study is of real value, it should provide a platform for further work both locally and nationally.

Reviewer 3 ·

Basic reporting

Thanks for your opportunity reviewing this paper. This study aims to assess associated factors with the survival status of IHCA patients.

Experimental design

1.Standards should be used in the study such as the Ulstein-style definitions including survival to discharge or 30 days. (https://doi.org/10.1016/j.resuscitation.2019.08.021)
2.In Method section and Table 4, all variables adjusted in Multi-variables logistic regression should be given.

Validity of the findings

3.Witnessed arrest is of no-significance (1.18, 0.35-4.00 ) in Multi-variables logistic regression, but main text lacks explanation. In theory, the variables entering the model are collinear, so there is a situation that was difficult to explain.

Additional comments

4.The conclusion does not quite match the results.

---

## Round 0.2 · Major Revisions

Dear authors,

Thank you for your re-submission. My decision is based on your potentially unintentional overlook on explaining or including a couple of aspects:

- the results should be descriptive and not a conclusion; that is left for the conclusion section (i think that was what the reviewer meant by stating "The conclusion does not quite match the results".

- the explanation about the validity of the findings is not clear and/nor it is included in the manuscript ("Witnessed arrest is of no-significance (1.18, 0.35-4.00 ) in Multi-variables logistic regression, but main text lacks explanation. In theory, the variables entering the model are collinear, so there is a situation that was difficult to explain.")

- a reviewer could not pinpoint the Utstein-style guidelines for reporting on in-hospital cardiac arrest. ("In main text, Ulstein-style methods were not found.") thus, i ask you to try to highlight the compliance with these guidelines, maybe even beyond the variables: Circumstances of the cardiac arrest, including location and initial rhythm; Details about resuscitation efforts, including the quality and timing of interventions; Documentation of outcomes, including return of spontaneous circulation (ROSC) and survival rates. (https://www.resuscitationjournal.com/article/S0300-9572(19)30582-9/fulltext , as indicated by the reviewer)

This should enhance the reliability and comparability of its findings within the context of pre-existing literature on cardiac arrest outcomes, thus respecting the principles laid out by the Utstein guidelines.

Note: relevant responses and/or clarifications should also be incorporated into the manuscript (if we had these doubts, so other readers can have them).
Many thanks.

·

Basic reporting

No comment

Experimental design

No comment

Validity of the findings

No comment

·

Basic reporting

Thank you for the revised paper, which significantly enhances the presentation of the study. Very minor points:

1. There are a small number of misspellings in the text.
2. Percentages should be presented as 9.5% not 9.45%

Experimental design

Good.

Validity of the findings

Good.

Additional comments

As above

Reviewer 3 ·

Basic reporting

Thanks for your opportunity re-reviewing this paper. The issue raised last time still exists.

Experimental design

Major comments:

1.In main text, Ulstein-style methods were not found.

Validity of the findings

Major comments:


2.Which variables are adjusted in Table 4?
3.Witnessed arrest is of no-significance (1.18, 0.35-4.00 ) in Multi-variables logistic regression, but main text lacks explanation. In theory, the variables entering the model are collinear, so there is a situation that was difficult to explain.

Additional comments

4.The conclusion is written as the result.

---

## Round 0.3 · accepted · Accept

Dear authors,
i am moving forward your manuscript and accepting it for publication.
Congratulations!

Please, be thorough in your proofreading. For example, figure 2 has 2 labels for the light green colour of the pie chart. Additionally, legends should be more complete, with e.g. key observations and/or noteworthy registered aspects or indications of what the readers are looking at. Make sure all tables and figures are properly cited in the main text (including supplemental).

·

Basic reporting

The revised document includes a number of changes which make it suitable for publication.

Experimental design

Adequate.

Validity of the findings

Appropriate.

Additional comments

Happy to recommend.